# Cryoprobe Placement Using Electromagnetic Navigation System (IMACTIS® CT-Navigation™) for Cryoablation Treatment of Upper Kidney Pole Lesions and Adrenal Metastases: Experience from a Single-Center, 4-Year Study

**DOI:** 10.3390/diagnostics14171963

**Published:** 2024-09-05

**Authors:** A. Michailidis, P. Kosmoliaptsis, G. Dimou, G. Mingou, S. Zlika, C. Giankoulof, S. Galanis, E. Petsatodis

**Affiliations:** Interventional Radiology Department, General Hospital of Thessaloniki “G.Papanikolaou”, 56429 Thessaloniki, Greecevpetsatodis@hotmail.com (E.P.)

**Keywords:** computer applications, interventional non-vascular, CT, image manipulation/reconstruction, 3D computer applications, cancer, metastases

## Abstract

The aim of this study is to evaluate the safety and efficacy of the use of the IMACTIS^®^ CT-Navigation™-electromagnetic navigation system (EMNS) in cryoablation CT-guided procedures under local anesthesia for the treatment of upper kidney pole and adrenal lesions. We conducted a retrospective analysis of patients with upper kidney pole lesions and adrenal metastases who underwent cryoablation using the IMACTIS-CT^®^-EMNS between January 2019 and April 2023. The EMNS was used to guide the placement of the cryoprobes with CT guidance under local anesthesia. The primary outcome was technical success, defined as the successful placement of the cryoprobes in the target lesion. A total of 31 patients were studied, of whom, 25 patients were treated with cryoablation for upper pole kidney masses, and 6 patients underwent the cryoablation of adrenal metastases during the study period. The mean age was 60 years (range, 36–82 years), and 21 patients were male. All the upper kidney pole lesions were renal cell carcinomas, and regarding adrenal metastases, the primary cancer sites were the lungs (*n* = 3), breast (*n* = 2), and the colon (*n* = 1). The median size of the lesions was 3,8 cm (range, 1.5–5 cm). All procedures were technically successful, with the cryoprobes accurately placed in the target lesions under CT guidance using the EMNS, avoiding the penetration of any other organs using an oblique trajectory. No major complications were reported, and local tumor control was achieved in all cases. Our initial experience using the EMNS for cryoprobe placement during CT-guided interventional procedures under local anesthesia for the cryoablation treatment of upper pole kidney lesions and adrenal metastases showed that it is safe and effective.

## 1. Introduction

Cryoablation has emerged as an alternative to surgery for the treatment of small tumors in various organs of the body, including the liver, kidney, and lungs.

Renal cell carcinomas (RCCs), especially those smaller than 4 cm (T1a) [1], as well as adrenal metastases in oligometastatic patients are established indications for local treatment [2,3,4,5,6,7].

Cryoablation has been shown to be a safe and effective treatment option [6,7]. Accurate placement of the cryoprobes is essential for complete ablation of the tumor while minimizing damage to surrounding tissue [8]. Various imaging modalities have been used to guide cryoprobe placement, including CT, ultrasound, and MRI. However, these modalities have limitations in terms of accuracy and real-time visualization [9,10]. Lesions located in the upper pole of the kidney or the adrenal gland can be challenging to treat as they require a very oblique trajectory in order to avoid puncturing the liver or the lung. The use of an electromagnetic navigation system (EMNS) for cryoprobe placement during CT-guided interventional procedures (IMACTIS^®^ CT-Navigation™) has been proposed as a potential solution to these limitations [9,10].

Cryoablation works by delivering extreme cold through cryoprobes directly into the tumor [11,12], causing ice crystal formation within the cells, which leads to cell death. This minimally invasive approach is particularly beneficial for patients who are not good candidates for surgery due to medical comorbidities or those who prefer a less invasive option [13]. The technique’s success is heavily dependent on the precise placement of the cryoprobes to ensure that the entire tumor is adequately targeted while sparing the surrounding healthy tissue.

Traditionally, imaging modalities such as CT, ultrasound, and MRI have been employed to guide the placement of cryoprobes. CT guidance is the most commonly used modality due to its ability to provide detailed anatomical images, but it requires repeated scans to confirm probe placement, which can increase radiation exposure. Ultrasound offers real-time imaging but is limited by its inability to visualize certain tissue types as well as the ice ball created during the cryoablation procedure and its dependency on the operator’s skill. MRI provides excellent soft tissue contrast but is time-consuming and expensive, and its real-time capabilities are limited [9,10,12].

The advent of electromagnetic navigation systems (EMNSs) represents a significant advancement in the field of image-guided interventions. EMNSs enhance the accuracy of cryoprobe placement by providing real-time tracking and navigation capabilities. This type of system integrates with CT imaging to allow for continuous visualization of the cryoprobes during the procedure. This integration helps reduce the number of control scans needed, thereby decreasing radiation exposure to the patient and medical staff. Furthermore, an EMNS can improve the efficiency of the procedure by shortening the overall time required for probe placement [6,9,10,11].

Also, the system can allow the operator to target lesions with very oblique trajectories without penetrating the lung or the liver, minimizing the risk of complications such as pneumothorax, bleeding, or tumor seeding to other organs [14].

The IMACTIS^®^ CT-Navigation™ system is one such EMNS designed for use in CT-guided interventional procedures. This system uses electromagnetic sensors attached to cryoprobes, which communicate with a navigation system to provide real-time feedback on the position and trajectory of the probes. This system has shown promise in enhancing the accuracy of probe placement and reducing procedural complications [12,13].

This single-center, four-year study aims to evaluate the effectiveness and safety of using the IMACTIS^®^ CT-Navigation™ system for cryoprobe placement in the cryoablation treatment of upper kidney pole lesions and adrenal metastases. Our center has adopted this technology with the goal of improving patient outcomes by ensuring more precise and reliable cryoprobe placement and increasing the safety of the procedure without putting the oncological outcome at risk. This study will review our procedural success rates, complication rates, and overall patient outcomes, providing valuable insights into the clinical benefits and potential challenges of integrating EMNSs into cryoablation procedures.

The integration of EMNSs in CT-guided cryoablation procedures represents a promising development in interventional radiology. By combining the detailed anatomical imaging capabilities of CT with the real-time navigation provided by EMNSs, we aim to achieve higher precision in probe placement, reduced procedural times, and minimized radiation exposure.

## 2. Materials and Methods

### 2.1. Study Design

This single-center, retrospective study analyzed patients with upper pole kidney lesions and adrenal metastases who underwent cryoablation using the IMACTIS^®^ CT-Navigation™ system from January 2019 to April 2023. Institutional Review Board approval was obtained for the study.

We included consecutive patients (Table 1) diagnosed with T1a and T1b renal cell carcinomas located in the upper pole of the kidney or by adrenal metastases, who received CT-guided cryoablation at our center during the study period.

In all patients, the treatment plan was established by a multidisciplinary tumor board consisting of urologists, abdominal surgeons, oncologists, and interventional radiologists.

All patients with metastatic cancer to the adrenal glands had already exhausted all available treatment options prior to the study, and therefore, no further follow-up treatments were undertaken.

The decision to perform the renal cryoablation of T1a RCCs was based on NCCN Guidelines Insights: Kidney Cancer, Version 2.2020 [3]. Also, cryoablation was performed on patients with T1b RCC and adrenal metastasis who were not eligible for surgery and were discussed at the tumor board.

The patients had renal or adrenal localization of the disease and, in some cases, other tumors that had been treated with no evidence of progression at the time of study inclusion, indicating stability either through prior intervention or natural disease course.

Life expectancy was estimated by an oncologic multidisciplinary team, taking into consideration factors such as patient comorbidities, overall performance status, and historical data on disease progression.

Statistical Analysis: Categorical variables were reported as frequencies and percentages, while continuous variables were summarized as medians with interquartile ranges (IQRs). Patients were stratified into two groups based on whether an EMNS was used during the procedure. Comparisons between these groups were performed using the Chi-Square test or Fisher’s Exact test for categorical variables.

Cryoprobe placement accuracy was defined as achieving positioning of the probe ± 5 mm from the pre procedural plan.

Mean procedural time was defined as the time in minutes from the application of the local anesthesia to the removal of all the probes and needles from the patient.

Cryoprobe placement time was defined as the mean time in minutes needed to correctly position each probe in the final position.

All statistical analyses were conducted using R software (version 3.5.0). Statistical tests were two-sided, with a significance level set at *p* < 0.05.

### 2.2. Pre-Procedure Preparation

Patients with adrenal metastases were given oral alpha-blockers before the procedure to mitigate the risk of hypertensive crisis during cryoablation starting 7 days prior to the procedure [15,16].

#### Procedure

Patients were positioned in the prone position for the procedure. Three interventional radiologists performed all procedures using a CT system for guidance. The cryoablation was carried out using a system that employed argon and helium gasses, with up to five cryoprobes, each 1.47 mm in diameter and capable of creating a freezing area of 1.5 × 3.5 cm.

Prior to cryoablation, a CT scan was performed to determine the tumor’s location, size, and extent. The number of cryoprobes used was based on the size of the tumor. Puncture sites and cryoprobe distribution were planned according to the tumor’s location, shape, and surrounding structures.

The IMACTIS^®^ CT-Navigation™ system (GE HealthCare Chicago, Heller International Building, 500 W Monroe St, Madison, WI, USA) operates with parameters set at 120 kV, an average of 281 mA, a pitch of 1.38, and a slice thickness of 1.25 mm. It comprises a touch-screen station and an electromagnetic locator, which includes a field generator placed near the puncture site for the automatic registration of magnetic and CT coordinates, and a magnetic receiver within the needle holder. Pre-procedure CT scans are swiftly uploaded to the navigation station, allowing the system to display the needle holder’s position and orientation on two perpendicular 2D images reconstructed from the 3D CT volume. This setup enables the radiologist to use the needle holder as a 3D mouse to explore the scan, locate the target, and plan the entry point and trajectory directly in the CT room. After sterile preparation, the needle is placed in the holder, and the radiologist inserts it when the optimal trajectory is displayed. The system provides real-time needle depth feedback by entering the needle length, and additional CT scans can be performed during insertion to monitor the needle’s progress, ensuring precise targeting.

All procedures were performed under local anesthesia. The adrenal cryoablation procedures were performed with arterial lines for blood pressure monitoring and the presence of an anesthesiologist, but still under local anesthesia [15,16].

### 2.3. Cryoablation Technique

The EMNS system was utilized to guide the precise placement of the cryoprobes under CT imaging. The edge of the freezing zone was maintained at 0 °C, with temperatures below −40 °C required to achieve cell destruction; thus, the freezing zone extended 0.5 to 1 cm beyond the tumor margin. The numbers of cryoprobes used depended on the size of the lesion. Cryoprobes were spaced 1–1.5 cm apart as per the instructions provided by the manufacturer for optimal spacing [11,12].

After verifying the puncture sites, local anesthesia was administered using 5 mL of 2% lidocaine. Following the placement of the cryoprobes, the tumor was subjected to a double freezing cycle protocol (10 min each cycle), with a 9 min interval of passive thawing and 1 min of active thawing.

The rapid expansion of argon gas within a sealed cryoprobe, featuring an uninsulated distal portion, led to the swift freezing of the tumor tissue, with cryoprobe tip temperatures reaching approximately −120 °C in seconds [8].

When there was proximity of the lesion to a sensitive structure or organ, a hydrodissection was performed to avoid any thermal damage. For the hydrodissection, a solution of 100 mL of normal saline mixed with 3 mL of contrast medium was used [14,17].

The ice ball formation, which represents the ablation zone, was monitored during the whole procedure with CT scans in order to ensure adequate coverage of the lesion with safety margins and also to detect any potential complications [18,19,20,21] (Figure 1).

### 2.4. Postoperative Management and Follow-Up

Following cryoablation, patients were observed for 30 min and then returned to the ward if they experienced no discomfort. Vital signs were closely monitored for the first 6 h post-treatment. Patients received appropriate anti-inflammatory and hemostasis treatments for 3 to 5 days following the procedure.

The follow-up period continued until the patient’s death or their last visit up to April 2023. Routine physical examinations and laboratory tests, including blood cell counts, adrenal hormone levels, and tumor markers based on primary tumor histology, were conducted monthly. Chest, abdominal, and pelvic CT scans with and without contrast enhancement were performed at 1, 3, 6, and 12 months post-cryoablation, and then every 6 months thereafter [11,12].

The primary outcome assessed was technical success, which was defined as the effective placement of cryoprobes within the target lesion. Secondary outcomes included the duration of the procedure, complications, and local tumor control. Technical success was determined by the completion of cryoablation following the planned treatment protocol and the absence of visible tumor enhancement on the initial contrast-enhanced CT or MR images obtained both at the end of treatment and 1 month afterward. The ablation zone in the surrounding fat tissue was challenging to detect on CT or MR images. If tumor enhancement persisted, the treatment was repeated after one month. Local tumor control was characterized by the absence of residual or new sites of tumor enhancement in follow-up imaging. Local tumor progression was defined as the emergence of a new enhancing renal or adrenal tumor during the follow-up period after achieving technical success with cryoablation. Systemic progression included local tumor progression, the development of new metastases, or the progression of the primary tumor (Figure 2).

## 3. Results

A total of 25 patients underwent the cryoablation of renal tumors and 6 patients underwent the cryoablation of adrenal metastases during the study period.

The demographic and clinical data of patients and lesions included in the present study are presented in Table 2. The mean age was 60 years (range, 36–82 years), and 19 patients were male. The primary cancer sites were the kidneys (*n* = 25), lung (*n* = 3), breast (*n* = 2) and colon (*n* = 1). The median size of the treated masses was 3.3 cm (range, 1.5–5 cm).

All procedures in this study were technically successful, with cryoprobes precisely placed in the target lesions under CT guidance using the electromagnetic navigation system (EMNS).

The tip position was estimated with a mean accuracy of 0.87 mm (standard deviation: 0.43 mm). The mean procedural time was 60 min (range: 40–80 min), and the mean cryoprobe placement time was 20 min (range: 15–25 min), as analyzed in Table 3.

Table 3 presents an analysis of cryoprobe placement accuracy, procedural times, and complication rates. Data are expressed as mean ± standard deviation.

Primary and secondary clinical success, defined as complete tumor ablation, was achieved in 100% of cases, as confirmed by follow-up imaging at three months post-procedure.

No recurrences were observed during the one-year follow-up period.

No additional surgical or systematic treatments were administered during the follow-up period in any of the patients.

Most patients experienced no major complications. Mild complications (grade 1 and 2 according to CIRSE) [20] were observed in three patients (11.4%), including self-limiting perinephric hematomas (two cases) and transient hematuria (one case). No patients required hospitalization or invasive intervention for these complications. To provide a comprehensive analysis of the efficacy and efficiency of EMNS-guided cryoablation, these results should be compared with outcomes from previous cryoablation methods.

Statistical comparisons between our findings and those from other studies can help validate the advantages offered by EMNSs, particularly in primary clinical success rates and complication rates.

We searched the literature to calculate the clinical success and complication rates of cryoablation for the treatment of renal tumors and adrenal metastasis without the use of a guidance system to provide a better clinical reference.

We searched the Cochrane Library, PubMed, Embase, CNKI databases, and Science databases, and the date was from the above database establishment to August 2023. We used the following search terms: “kidney neoplasms”, “renal tumor”, “cancer of the Kidney”, ”adrenal metastasis”, “cryoablation”, “cryosurgery”, “clinical success”, and “complications”. Search strategies were tailored for the different search engines. Manual retrieval from the references of subject-related articles was performed to broaden the search. The search was not limited by region or language.

We included 11 studies [21,22,23,24,25,26,27,28,29,30,31] that matched our criteria and calculated the clinical success rates, which ranged from 83% to 99%, with a mean rate of 91% ± 0.5 (SD). We also calculated the mean minor complication rates to be between 5% and 15%, with a mean rate of 10% ± 0.1 (SD) (Figure 3).

Clinical Success Rates: Our study reported a 100% technical success rate with complete local tumor control, aligning with or exceeding the success rates reported in previous studies, which ranged from 83% to 99%.Complication Rates: Our study observed no major complications, as in most previous studies; we reported minor complications at a rate of 9.6%, consistent with previous studies, which reported minor complication rates between 5% and 15%.

**Figure 3 diagnostics-14-01963-f003:**
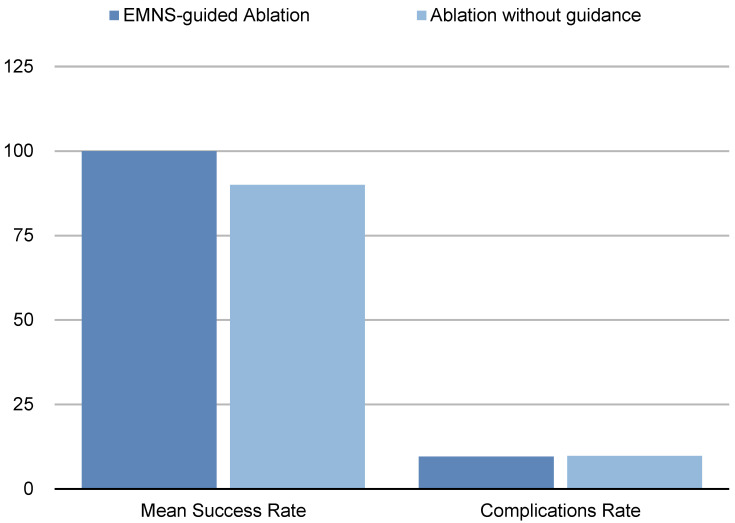
Bar plots comparing the primary clinical success rates and minor complication rates of EMNS-guided cryoablation with those from previous techniques.

These comparisons underscore the possible superior precision and efficiency and consistent safety of EMNS-guided cryoablation compared to traditional methods. The data suggest that EMNSs may significantly enhance procedural outcomes, offering a viable option for achieving optimal treatment results with fewer complications and shorter procedure times (although this small cohort of patients could not be used to successfully establish this hypothesis). Further studies comparing EMNSs with other technologies are recommended to solidify these findings and explore the long-term benefits of EMNS-guided cryoablation (Figure 4, Figure 5 and Figure 6).

## 4. Discussion

Cryoablation is an effective treatment for patients with T1a and T1b renal cell carcinoma (RCC) who are not ideal candidates for surgery due to comorbidities or tumor location. This minimally invasive procedure uses extreme cold to destroy cancer cells and is particularly indicated for tumors smaller than 4 cm (T1a) and also those between 4 and 7 cm (T1b) [23,24]. Additionally, cryoablation is a viable option for adrenal metastases, providing symptom relief and local control of the disease. Cryoablation is particularly advantageous for patients who are not ideal candidates for traditional surgical resection due to comorbidities or the location of the tumor. Its minimally invasive nature allows for quicker recovery times, less postoperative pain, and reduced risks of complications compared to open surgery. The precision of cryoablation also makes it a suitable option for treating tumors in delicate or hard-to-reach areas, such as the adrenal glands [22] and the upper pole of the kidney, where surrounding vital structures must be preserved.

However, while cryoablation is increasingly recognized as a valuable treatment option, particularly for small, localized tumors, it is not yet considered the standard treatment for metastatic cancer [26]. The standard of care for metastatic cancer typically involves a combination of systemic therapies, such as chemotherapy, targeted therapy, immunotherapy, and sometimes radiotherapy, depending on the type and extent of the disease. Surgical resection remains the preferred treatment for many primary tumors and selected metastatic lesions, particularly when complete removal is feasible and offers the best chance for long-term survival.

Cryoablation is most commonly used for patients with small, isolated metastatic lesions where systemic therapy alone may not be sufficient or for those who seek a less invasive alternative to surgery. It is also an option for patients with recurrent tumors or for those who cannot undergo surgery due to poor health or the high risk of surgical complications. Moreover, cryoablation is sometimes used as a palliative treatment to reduce the size of tumors and alleviate symptoms in advanced cases of metastatic cancer.

The treatment of tumors that have metastasized to the adrenal glands generally involves chemotherapy and radiation therapy tailored to the specific type of tumor. The role of surgery for local control remains debated. Nonetheless, patients with isolated metastatic disease confined to an adrenal gland have shown improved survival rates following adrenalectomy. Reports from the literature indicate that adrenal gland ablation has been employed for managing various conditions, including nonfunctioning adenomas, functioning adenomas such as cortisol-producing adenomas, aldosteronomas, adrenocortical carcinoma, pheochromocytomas, and metastases originating from renal cell carcinoma, melanoma, as well as lung and gastrointestinal cancers [22,24].

In recent years, EMNSs have been developed to aid in the placement of cryoprobes during CT-guided interventional procedures.

The use of an electromagnetic navigation system, such as the IMACTIS^®^ CT-Navigation™, enhances precision in cryoprobe placement, contributing to improved outcomes and reduced complications in these delicate procedures.

The accurate placement of the cryoprobes is crucial for the success of cryoablation in the treatment of adrenal metastases and upper kidney pole lesions. The use of an electromagnetic navigation system (EMNS) for cryoprobe placement during CT-guided interventional procedures offers several advantages over other imaging modalities. The EMNS allows for real-time visualization of the needle trajectory and accurate placement of the cryoprobes within the target lesion. This can help to reduce the risk of damage to surrounding organs and improve the efficacy of the procedure. Cryoablation is a minimally invasive technique for the treatment of adrenal metastases and upper kidney pole lesions, which involves the use of cryoprobes to freeze and destroy the tumor cells.

The integration of EMNSs in CT-guided procedures represents a significant advancement in interventional radiology. One of the primary benefits of using EMNSs is the reduction in radiation exposure for both the patient and medical staff. Traditional CT-guided procedures often require multiple scans to confirm needle placement, which can result in higher cumulative radiation doses. EMNSs can minimize the need for repeated imaging by providing real-time feedback on the position and trajectory of the cryoprobe. This not only reduces radiation exposure but also shortens the overall procedure time, which is advantageous in a clinical setting where efficiency is paramount.

However, these systems also have limitations. A notable disadvantage is their inability to account for uncontrolled patient breathing, which can cause organ movement and reduce targeting accuracy. This limitation necessitates additional measures, such as breath-hold commands or respiratory gating, to mitigate movement and ensure precise probe placement.

A study by N. Kelekis et al. (2021) [32] evaluated the use of IMACTIS^®^ CT-Navigation™ for probe placement during CT-guided interventional procedures for the treatment of malignant liver lesions. The prospective descriptive study included 10 patients who underwent microwave ablation treatment, with probe placement guided by the EMNS. The authors reported a primary technical success rate of 93.75% and a secondary technical success rate of 100%, with accurate placement of the cryoprobes achieved in all cases. This high success rate underscores the reliability of EMNSs in ensuring precise probe placement, which is crucial for the effective ablation of tumors.

Another study by H. Rabeh and F. Wacker et al. (2021) [33] compared the use of IMACTIS^®^ CT-Navigation™ with conventional CT guidance for probe placement during the percutaneous microwave ablation of malignant liver tumors. The study included 34 patients, with 17 patients undergoing probe placement guided by the EMNS and 17 patients undergoing conventional CT-guided probe placement. The authors reported a mean total deviation of the antenna feed point in the navigation and control group of 2.4 mm (range 0.2–4.8 mm) and 3.9 mm (range 0.4–7.8 mm), *p*  <  0.05. The mean setup time for the EMNS was 6.75  ±  3.9 min (range 3–12 min). The mean number of control scans in the navigation and control group was 3 ± 0.9 (range 1–4) and 6  ±  1.3 (range 4–8), *p*  <  0.0001; the mean time for antenna placement was 9  ±  7.3 min (range 1.4–25.9 min) and 11.45 ± 6.1 min (range 3.9–27.4 min), *p*  =  0.3164. Radiation exposure was significantly less in the navigation group. These findings indicate that EMNSs not only improve the accuracy of probe placement but also enhance the overall safety and efficiency of the procedure.

A study by S. Volpi et al. (2019) [34] evaluated the use of IMACTIS^®^ CT-Navigation™ included 27 patients who underwent thermal ablation treatment, with probe placement guided by the EMNS. The authors reported complete ablation was obtained in 100% at the 3- and 6-month MRI follow-up. The ablation probe was correctly placed on the first pass in 96%, with a mean path-to-tumor angle of 7  ±  4 degrees and a mean tip-to-tumor distance of 22  ±  19mm. A readjustment for additional overlapping application resulted in complete treatment in four patients. Needle placement took a mean 23  ±  12 min with mean radiation doses of 558 mGy*cm. No major complications were reported. This study further confirms the high accuracy and safety profile of EMNSs in guiding cryoprobe placement.

Overall, the literature suggests that the use of an EMNS for probe placement during CT-guided interventional procedures is safe and effective, with accurate placement of the probes achieved in all cases. The use of an EMNS may also reduce the procedural time and improve the efficacy of the procedure, which are results that are also in concurrence with the results of our study.

However, most of these studies were performed for microwave ablation techniques and especially liver ablation [32,33,34], where mostly one needle probe was used and manual positioning was already not that time-consuming compared to cryoablation, where it usually needs at least two probes and the placement of the probe needs to have a high level of accuracy.

This is why our study, describing our initial experience using the IMACTIS^®^ CT-Navigation™ system for cryoprobe placement during CT-guided interventional procedures for the cryoablation treatment of adrenal metastases and upper kidney pole lesions, was aiming to explore the importance of navigational tools and especially EMNSs in complex cases that are more challenging to the interventionalist.

Future research should focus on larger multi-center studies to further validate these findings and explore the long-term outcomes of patients treated with EMNS-guided cryoablation. Additionally, investigations into the cost-effectiveness of EMNS in various clinical settings could provide further justification for its widespread adoption.

## 5. Conclusions

The use of an electromagnetic navigation system (EMNS), specifically the IMACTIS^®^ CT-Navigation™, for cryoprobe placement during CT-guided interventional procedures for the cryoablation treatment of upper kidney pole lesions and adrenal metastases has proven to be both safe and effective [35,36]. In our single-center, four-year experience, accurate placement of the cryoprobes was achieved in all cases, demonstrating the reliability and precision of the EMNS. The system’s real-time visualization capabilities significantly enhance the operator’s ability to monitor and adjust the needle trajectory, thereby reducing the risk of damage to surrounding tissues and improving the overall efficacy of the procedure.

The integration of EMNSs in these procedures offers several key advantages. The ability to continuously track the position of the cryoprobes in real time not only enhances the accuracy of probe placement but also reduces the need for multiple confirmatory scans, thereby decreasing radiation exposure for both patients and medical staff. Additionally, the system streamlines the procedural workflow, potentially reducing the total procedure time and associated healthcare costs.

An EMNS is a guiding system that helps with the correct positioning of the needle which is paramount to achieve the complete ablation of a tumor and ablation margins that are safe. Good primary and secondary results can be achieved through this method. Without this relative new technology, it was difficult to achieve complete ablation, especially in upper kidney pole masses and adrenal masses.

Despite the promising outcomes observed in our study, it is important to acknowledge the need for further research to confirm these findings in larger and more diverse patient populations. Larger studies will help to validate the generalizability of our results and provide more comprehensive data on the long-term outcomes and potential complications associated with EMNS-guided cryoablation. Furthermore, comparative studies between different EMNS technologies could offer valuable insights into their respective advantages and limitations, guiding clinicians in selecting the most appropriate system for their specific clinical needs.

Our experience suggests that the adoption of EMNSs for cryoablation procedures can set new standards for precision and safety in minimally invasive tumor treatments. As technology continues to evolve, future innovations in EMNS and image-guided interventions will likely further enhance the effectiveness and accessibility of cryoablation and other interventional radiology procedures. This study contributes to the growing body of evidence supporting the use of EMNSs in clinical practice and highlights the potential benefits of integrating advanced navigation systems in the treatment of complex lesions.

In conclusion, the IMACTIS^®^ CT-Navigation™ system has demonstrated significant potential in improving the accuracy and safety of cryoprobe placement for the cryoablation of upper kidney pole lesions and adrenal metastases. While our findings are encouraging, ongoing research and clinical trials will be essential to fully realize the benefits of EMNS in interventional radiology and ensure optimal patient outcomes.

## Figures and Tables

**Figure 1 diagnostics-14-01963-f001:**
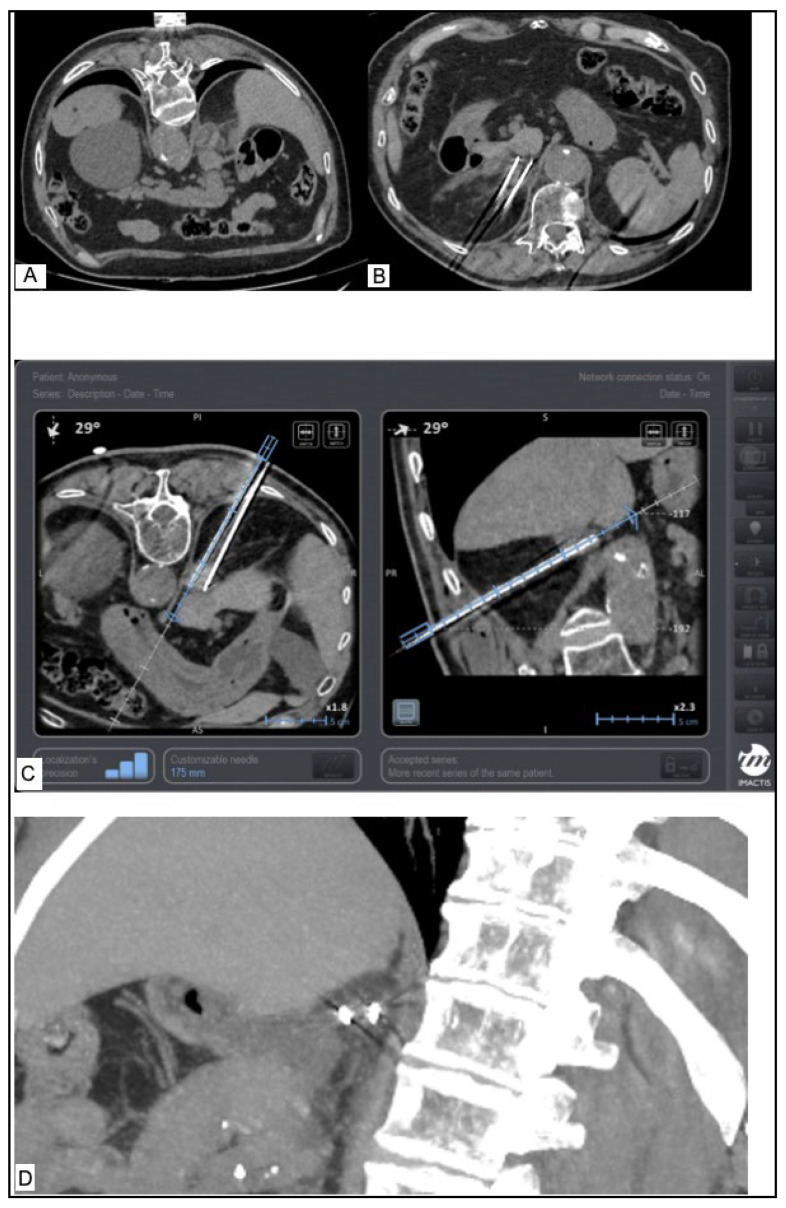
65-year-old male patient with lung adenocarcinoma and adrenal metastasis. (**A**) Axial CT image shows mass in right adrenal gland, diameter 3 cm. (**B**) Axial CT image shows ground glass opacity in the periphery of the mass but no other complication. (**C**) using the EMNS system showing the placement of cryoprobes in the correct position in the adrenal mass. (**D**) CT after starting freezing process picturing the ice ball (ablation zone) covering the whole mass.

**Figure 2 diagnostics-14-01963-f002:**
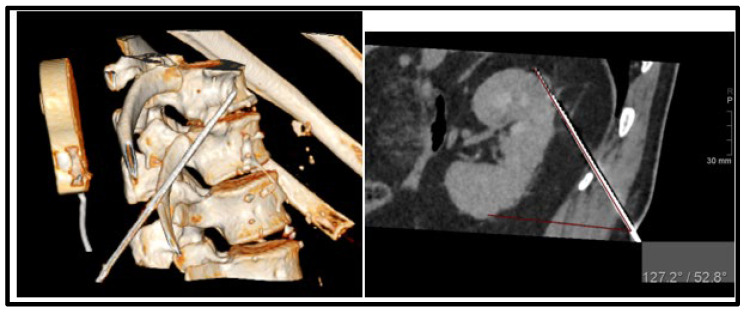
Cryoprobe placement using the EMNS with a steep angle of 52–3D rendering and sagittal CT image.

**Figure 4 diagnostics-14-01963-f004:**
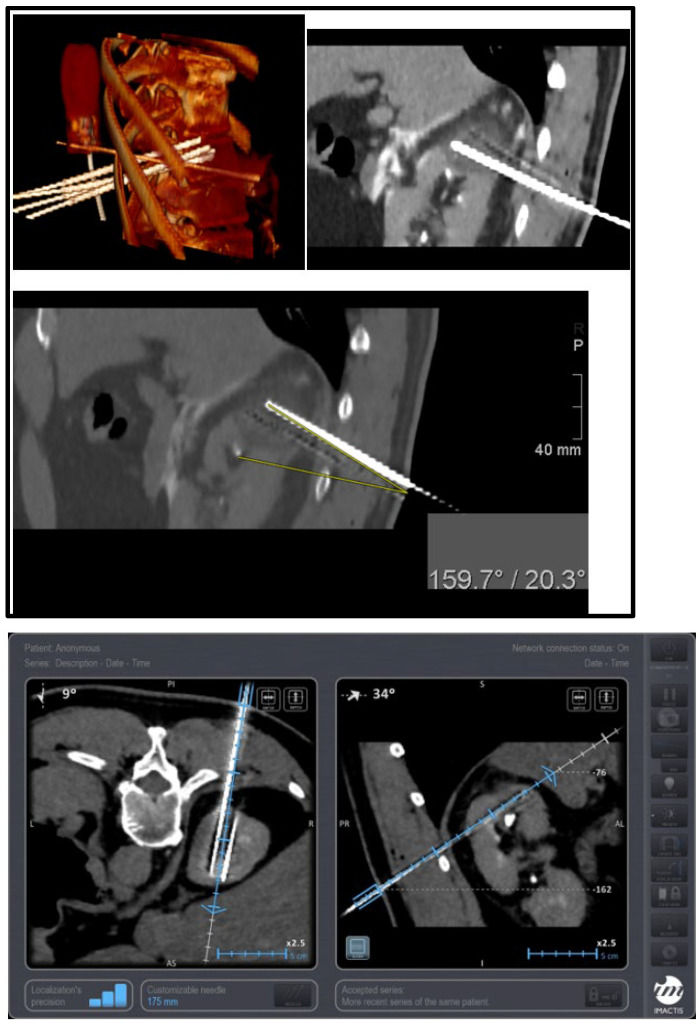
Cryoablation of upper pole renal mass—3D rendering depicting the four cryoprobes that were placed with a 1–1.5 cm distance between them—sagittal CT image placement with an EMNS with a minimum angle of 20.

**Figure 5 diagnostics-14-01963-f005:**
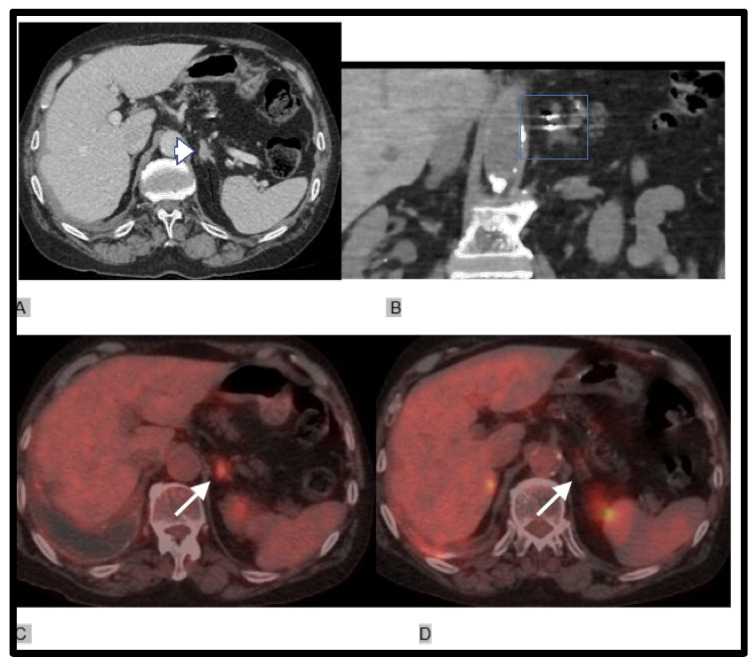
(**A**) 75-year-old male patient with stage IV lung adenocarcinoma with bilateral metachronous adrenal metastasis. After immunotherapy cryoablation of the single lung nodular mass. Mass of 2.5 cm (arrow head) in left adrenal gland. (**B**) CT before starting the freeze cycles showing the placement of cryoprobes (square) in the correct position using IMACTIS^®^ CT-Navigation™in the adrenal mass. (**C**,**D**) The adrenal lesion (white arrow) before and after successful cryoablation; there is no residual FDG uptake on PET/CT, and the patient is cancer-free 1 year after.

**Figure 6 diagnostics-14-01963-f006:**
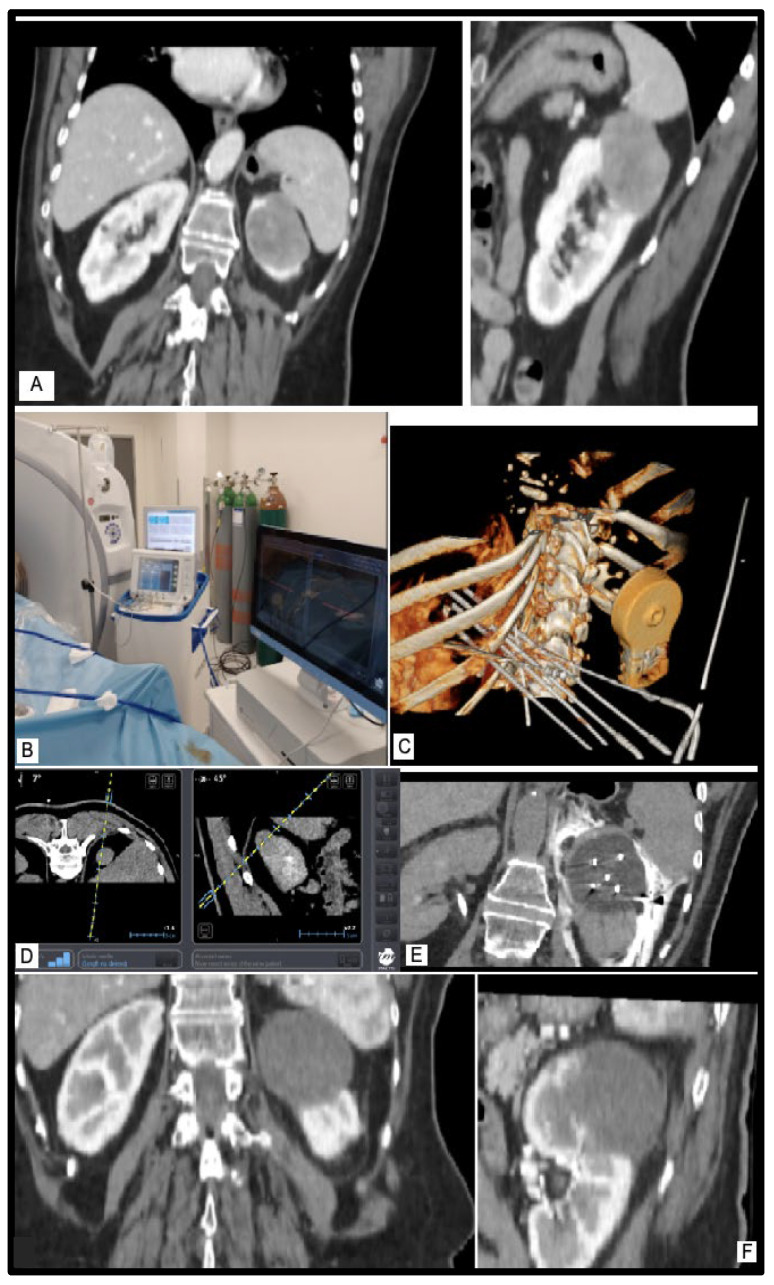
Example of technically successful T1b cryoablation as demonstrated by (**A**) pre-ablation coronal and sagittal contrast-enhanced CT image depicting exophytic T1b RCC of the upper pole of the left kidney (maximum diameter 48 mm). (**B**) IMACTIS^®^ CT-Navigation™ monitor next to cryoablation generator by the patient’s side. (**C**) Three-dimensional rendering of the final needle placement and the fiducial device. (**D**) Active needle tracking with pseudo axial and sagittal image of the lesion. (**E**) Final placement of the cryoprobes with optimal spacing, ice ball formation, and hydrodissection. (**F**) Ablation zone immediately post-procedure demonstrating ice ball with complete tumor coverage and appropriate margin.

**Table 1 diagnostics-14-01963-t001:** Inclusion and exclusion criteria.

No	Inclusion Criteria	Exclusion Criteria
1	Tumor size less than 6 cm.	Primary adrenal neoplasms.
2	T1a and TIb RCCs.	Patients with invasion of the renal vein.
3	No or controlled extra-adrenal or kidney tumors.	Significant blood coagulation disorders, active infections, or active bleeding.
4	Life expectancy of at least 6 months.	
5	Patients not eligible for surgical intervention or who refused surgery.	

**Table 2 diagnostics-14-01963-t002:** Demographic and clinical data of the selected patients and lesion before cryoablation.

Baseline Data of the 6 Patients.	Value
Patients number	31
Sex, male/female	19/12
Age, years	57.9 ± 9.6
Primary disease	Renal cancer, 25; lung cancer, 3; breast cancer, 2; colonic cancer, 1
Extent of disease at time of cryoablation	Isolated renal tumor, 25; isolated adrenal metastasis, 6; renal tumor and adrenal metastasis, 0; metastases in adrenal gland and other sites, 0
Previous surgical resection	Yes, 5; no, 26
Previous systemic therapy	Yes, 4; no, 27
Systemic therapy after cryoablation	Yes, 0; no, 31
Tumor size, cm	3.3 ± 0.7
Side	Left, 12; right, 19

**Table 3 diagnostics-14-01963-t003:** Analysis of cryoablation outcomes.

Parameter	Metric	Results
Cryoprobe Placement Accuracy	Mean Accuracy (mm)	0.87
Mean Procedural Time	Time (minutes)	60
Cryoprobe Placement Time	Time (minutes)	20
Complication Rates	Major Complications (%)	0
Complication Rates	Minor Complications (%)	9.6

## Data Availability

The original contributions presented in the study are included in the article, further inquiries can be directed to the corresponding author.

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
