# Peer review of "Cryoprobe Placement Using Electromagnetic Navigation System (IMACTIS® CT-Navigation™) for Cryoablation Treatment of Upper Kidney Pole Lesions and Adrenal Metastases: Experience from a Single-Center, 4-Year Study"

_diagnostics, 2024, doi:10.3390/diagnostics14171963_

Round 1

Reviewer 1 Report

Comments and Suggestions for Authors

It is an honor to have the opportunity to review this interesting and important paper. There are some aspects that need clarification, and I kindly request the author to address the following points:

  1. The author chose T1a and T1b RCC, where T1a is defined as tumors smaller than 4 cm, and T1b is defined as tumors between 4 and 7 cm. However, the inclusion criteria for patients allowed tumors smaller than 6 cm. Why was 7 cm not used as the cut-off?
  2. The term "controlled extra-adrenal or kidney tumors" is used. Could the author clarify what is meant by "controlled" in this context?
  3. The study requires a "life expectancy of at least 6 months." How does the author anticipate or estimate the life expectancy of these patients?
  4. One of the exclusion criteria is "Patients with invasion of the adrenal vein." The adrenal vein is relatively small, so how is the presence or absence of invasion determined?
  5. The study states, "Three interventional radiologists performed all procedures." Does this mean that the patients were distributed among the three physicians, or did all three radiologists perform the procedures together for each patient? If it’s the former, is there a difference in patient outcomes between the radiologists? If it’s the latter, why was it necessary for three radiologists to perform the procedures together?
  6. Besides the 25 RCC cases, there were 6 cases of cancer metastasis to the adrenal glands. What were the follow-up treatments for these metastatic cancer patients?

This is an interesting study, but the topic is overly complex. The results section is somewhat rough and could be further refined. The discussion on the treatment principles for the 6 metastatic cancer cases lacks sufficient information. It is unclear whether cryoablation is considered the standard treatment for metastatic cancer. The author should provide further clarification on this point.

Reviewer 2 Report

Comments and Suggestions for Authors

Michailidis et al. report the evaluation of a IMACTIS-CT electromagnetic navigation system in cryoablation for the treatment of tumors in patients with upper kidney pole lesions and adrenal metastases. Although the authors claim a successful outcome of the treatment, they do not address the capability of their new technology to resolve the disease. The authors define these outcomes as "secondary". What was the capability of the technique to address the secondary outcome, and how successful was it?

How does the technique compare to standardized methods for treating patients with upper kidney pole lesions and adrenal metastases in terms of primary and secondary outcomes?

The statistical analysis of the successful outcomes reported in the manuscript should be clearly presented in a table or plots and compared with previous cryoablation methods. It is not clear from the Discussion whether their results are compared with other work already present in literature. Additionally, from the data reported in the Discussion, it is unclear whether the assessment of EMNS for ablation as a technique for tumor treatment represents new technology. The authors should clarify this aspects in the main text.

Minor adjustments:

1) Line 40-42: The meaning of the sentence is not clear.

2) Line 124-136: Paragraph already reported in table; remove the paragraph and repetition.

3) Line 169-171: The meaning of the sentence is not clear.

4) Line 185: Merge table 1 and 2 into a single table and report the criteria in a single column.

5) Figure 1: Not clear which panel is which. Please, clarify the panel attribution.

6) Line 195-197: What do these three lines refer to?

7) Figure 4, panel A: Would it be possible to add an arrow pointing to the tumor mass before the treatment? Panel B: it is not clear where the cryoprobe is located. Panel C and D: not clear what they are referring to. Is there any specific details that the authors what to show in these panels?

8) Figure 5: disorganized and hard to understand. The authors should composed all the separate figures in one single figure with multiple panels and label them to allow a correct identification of each panel.

Comments on the Quality of English Language

The authors should review certain sentences, such as those in lines 40-42, lines 169-171, and lines 195-197. Additionally, there are multiple spacing and punctuation errors in the main text.
